# Effects of the Chloroplast Fructose-1,6-Bisphosphate Aldolase Gene on Growth and Low-Temperature Tolerance of Tomato

**DOI:** 10.3390/ijms23020728

**Published:** 2022-01-10

**Authors:** Bingbing Cai, Yu Ning, Qiang Li, Qingyun Li, Xizhen Ai

**Affiliations:** 1Horticulture Department, Hebei Agriculture University, Baoding 071000, China; caibingbing@hebau.edu.cn (B.C.); yylq@hebau.edu.cn (Q.L.); yylqy@hebau.edu.cn (Q.L.); 2Horticulture Department, Shandong Agriculture University, Tai’an 271018, China; ny271565853@126.com; 3State Key Laboratory of North China Crop Improvement and Regulation, Hebei Agricultural University, Baoding 071001, China; 4Institute of Vegetables and Flowers, Chinese Academy of Agricultural Sciences, Beijing 100081, China

**Keywords:** tomato, fructose-1,6-bisphosphate aldolase, transgenic, low-temperature stress, photosynthesis, growth

## Abstract

Tomato (*Solanum lycopersicum*) is one of the most important greenhouse vegetables, with a large cultivated area across the world. However, in northern China, tomato plants often suffer from low-temperature stress in solar greenhouse cultivation, which affects plant growth and development and results in economic losses. We previously found that a chloroplast aldolase gene in tomato, *SlFBA4*, plays an important role in the Calvin-Benson cycle (CBC), and its expression level and activity can be significantly altered when subjected to low-temperature stress. To further study the function of *SlFBA4* in the photosynthesis and chilling tolerance of tomato, we obtained transgenic tomato plants by the over-expression and RNA interference (RNAi) of *SlFBA4*. The over-expression of *SlFBA4* led to higher fructose-1,6-bisphosphate aldolase activity, net photosynthetic rate (Pn) and activity of other enzymes in the CBC than wild type. Opposite results were observed in the RNAi lines. Moreover, an increase in thousand-seed weight, plant height, stem diameter and germination rate in optimal and sub-optimal temperatures was observed in the over-expression lines, while opposite effects were observed in the RNAi lines. Furthermore, over-expression of *SlFBA4* increased Pn and enzyme activity and decreased malonaldehyde (MDA) content under chilling conditions. On the other hand, Pn and MDA content were more severely influenced by chilling stress in the RNAi lines. These results indicate that *SlFBA4* plays an important role in tomato growth and tolerance to chilling stress.

## 1. Introduction

Fructose-1,6-bisphosphate aldolase (FBA, EC 4.1.2.13) has been extensively found in various organisms, such as bacteria, higher plants and animals [1,2]. It is an essential enzyme that is involved in carbohydrate metabolism, including gluconeogenesis, glycolysis and the Calvin-Benson cycle (CBC) [3,4]. In the CBC, FBA catalyzes the reversible conversion of fructose-1,6-bisphosphate (FBP) to dihydroxyacetone phosphate (DHAP) and glyceraldehyde-3-phosphate (G3P) [5,6,7]. Both DHAP and G3P are substrates or metabolic products in the tricarboxylic acid cycle, phenylpropanoid pathway and oxidative/non-oxidative pentose phosphate pathway [8].

FBAs can be divided into two classes based on their catalytic mechanisms and evolutionary origins [2]. Class I FBAs utilize a lysine residue to generate a Schiff base, intermediate product, and their activity can be inhibited by hydroboron [2]. Class I FBAs often occur in higher plants and animals and have two isoforms that differ in subcellular locations: chloroplast/plastid FBA (CpFBA) and cytosolic FBA (cFBA) [9,10]. CpFBA catalyzes the condensation reaction of FBP and sedoheptulose-1,7-bisphosphate [11], whereas cFBA plays a vital role in glycolysis and gluconeogenesis [9,12]. Class II FBAs are mainly identified in microorganisms [5,11] and only a few class II FBAs occur in wheat and other closely related species [13]. Class I FBAs have been extensively identified and characterized in various plants, such as *Arabidopsis* [10], rice [14,15], maize [16], spinach [9], soybean [17], potato [18], oat [19], tobacco [20], *Sesuvium portulacastrum* L. [21], bamboo [22], wheat [13] and tomato [23,24]. Many studies have indicated that FBAs are involved in plant growth and development [7,20]. For example, higher cFBA activity was observed in elongating tissues than in tissues that had finished elongating in moso bamboo (*Phyllostachys pubescens* Mazel), which indicated an important role of cFBA in the elongation of tissues [22]. In tomato, decreased FBA activity resulted in the slower growth of plants [23,24]. Accumulating evidence indicates that FBAs respond to various biotic [8] and abiotic stresses [10,19,20,21,24,25,26,27,28]. For example, FBA activity was up-regulated in 12-day-old wheat seedlings under salt stress, which helped the seedlings adapt to the stress [26]. In oat (*Avena sativa* L.), FBA activity can be induced by heat [19], whereas in chickpea (*Cicer arietinum* L.), FBA activity was repressed by water-deficit stress [28].

We previously identified eight members of the *FBA* gene family in tomato, which comprised five *CpFBA*s and three *cFBA*s. *SlFBA7*, one of the *cFBA*s, plays an important role in regulating tomato growth and chilling tolerance [24]. However, little is known about the function of *CpFBA*s in tomato. In the present study, we focus on *SlFBA4*, one of the *CpFBA*s, to investigate the function of this gene and demonstrate that it also plays an important role in regulating tomato growth and chilling tolerance.

## 2. Results

### 2.1. Production and Selection of Tomato Transformants

To test whether *SlFBA4* regulates plant growth, especially under low temperature, an over-expression cassette that contained the *SlFBA4* CDS controlled by the CaMV-35S promoter (*35S*-*SlFBA4*) and an RNAi vector that contained a reverse-complementary hairpin structure (*RNAi*-*SlFBA4*) were created and introduced into tomato via *Agrobacterium tumefaciens*-mediated genetic transformation (Appendix A). Transgenic plants that overexpressed and suppressed *SlFBA4* were generated. Three independent over-expression lines (T2-19, T2-22 and T2-33) and RNAi lines (T2′-16, T2′-30 and T2′-31) with significantly increased or decreased *SlFBA4* expression were used for further investigation (Figure 1). Compared with *SlFBA4* expression in wild-type (WT) plants, *SlFBA4* transcript abundance in the three over-expression transgenic lines increased by 178.39%, 208.09% and 1480.27% (Figure 1A). The *SlFBA4* expression level of the three RNAi lines was decreased by 35.93%, 69.27% and 96.37% (Figure 1D). Meanwhile, the FBA activities of the *SlFBA4* over-expression lines were significantly increased compared with WT (Figure 1B). In contrast, the RNAi lines exhibited significantly decreased FBA activities when compared with WT (Figure 1E). Similarly, stronger and weaker protein signals were observed in the over-expression and RNAi transgenic lines, respectively, than that in WT plants using Western blot analysis with an antiserum against SlFBA4. 

### 2.2. Effects of SlFBA4 on Transgenic Tomato Plant Growth

To investigate the effects of *SlFBA4* on tomato seed and seedling development, the thousand-seed weight, germination rate and plant growth parameters, including plant height and stem diameter, were measured. As shown in Table 1, over-expression of *SlFBA4* significantly increased the thousand-seed weight compared with that in WT. Conversely, the down-regulation of *SlFBA4* decreased the thousand-seed weight compared with that in WT. Under 28 °C, which is the normal temperature for tomato seed germination, the difference in germination rate between WT and over-expressed transgenic seeds was not obvious; however, the germination rate and germination potential of RNAi transgenic seeds was significantly lower than that of WT (Table 1; Figure 2). For plant growth, over-expression of *SlFBA4* resulted in plants with a significantly larger plant height and stem diameter than those of WT (Table 1; Figure 2). However, plant height and stem diameter were comparable between the RNAi plants and WT (Table 1; Figure 2). 

### 2.3. Effects of SlFBA4 on Net Photosynthetic Rate as well as Gene Expression and Activity of Select Enzymes in the Calvin-Benson Cycle

Since CpFBA is a key enzyme in the CBC, which is an important stage of photosynthesis, we also measured Pn in transgenic tomato plants. As expected, Pn significantly increased and decreased in the *SlFBA4* over-expression lines and the RNAi lines, respectively, compared with that in WT (Figure 3), which indicated an important role of *SlFBA4* in regulating photosynthesis in tomato.

Previously, we found that the down-regulation of tomato *FBA7* led to significant changes in the mRNA expression and activity of some enzymes in the CBC, including ibulose 1,5-bisphosphate carboxylase/oxygenase (*Rubisco*), sedoheptulose-1,7-bisphosphatase (*SBPase*), fructose 1,6-bisphosphatsse (*FBPase*), glyceraldehyde-3-phosphate dehydrogenase (*GAPDH*) and transketolase (*TK*). Therefore, we investigated the effects of *SlFBA4* on the mRNA expression and activity of these enzymes. As shown in Figure 4, the relative expressions of *rbcL*, *rbcS*, *SBPase*, *FBPase*, *GAPDH* and *TK* were significantly increased in the over-expression lines compared with that in WT, except *TK* in lines T2-19 (Figure 4). However, a significant decrease in the mRNA abundances of the six genes in the RNAi lines was found when compared with WT (Figure 5). As expected, the activities of RuBPCase, SBPase, FBPase, GAPDH and TK were significantly increased in at least two over-expression lines compared with that in WT (Figure 6). Conversely, significantly lower activities of the five enzymes were observed in the RNAi lines than in WT (Figure 5 and Figure 7). Taken together, these results suggest an important role of *SlFBA4* in regulating photosynthesis in tomato.

### 2.4. Effects of SlFBA4 on Germination Rate, Net Photosynthetic Rate and Malonaldehyde Content under Chilling Stress

To explore the effects of *SlFBA4* on chilling tolerance in tomato, germination rate, Pn and MDA content were investigated in WT and transgenic seedlings under chilling conditions. Under 18 °C, which is a sub-low temperature for tomato seed germination, germination rate increased and decreased in *SlFBA4*-over-expressing and RNAi lines, respectively, compared with that in WT (Table 1). Similar changes in Pn were also observed in transgenic lines compared with that in WT under 5 °C treatment for 48 h (Figure 8A,B). MDA is the main product of lipid peroxidation and its content can indicate damage to plant membranes. As shown in Figure 8C,D, MDA content in WT and over-expression seedlings T2-19, T2-22 and T2-33 under chilling for two days significantly increased by 129.33%, 89.19%, 88.51% and 102.08%, respectively, compared with that under normal conditions. MDA content in RNAi seedlings T2′-16, T2′-30 and T2′-31 increased by 201.32%, 202.63% and 230.34%, respectively, compared with that under normal conditions. These results indicate that the over-expression of *SlFBA4* increased chilling tolerance and interference of *SlFBA4* expression increased damage to the cell membranes of tomato seedlings by chilling stress.

## 3. Discussion 

Tomato is a vegetable crop cultivated worldwide. In northern China, tomato is widely cultivated in greenhouses during winter and spring. Thus, tomato plants often suffer from low-temperature stress, leading to reduced productivity and economic losses. Chilling stress is a common factor that leads to a dramatic reduction in photosynthesis in chilling-sensitive plants, with the consequence of damaging the photosynthetic apparatus [29,30].

It has long been known that the limitations of carbon assimilation in C3 plants are largely due to the catalytic properties of ribulose-1,5-bisphosphate carboxylase/oxygenase (Rubisco) [31]. However, since the 1990s, studies have found that reductions in the activity of Rubisco have little impact on carbon assimilation under normal environmental conditions [32,33,34]. Rubisco may not be the primary limiting factor that leads to poor performance under chilling conditions but instead helps plants recover faster from these conditions [35]. Recent studies have shown that there is a non-regulated enzyme, aldolase, that is sensitive to abiotic stresses [36,37,38]. It is well known that FBAs are key positive regulators of photosynthesis [39,40]; however, the knowledge of SlFBAs in regulating photosynthesis and chilling stress in tomato is still limited. 

We previously identified eight FBA genes in the tomato genome, which can be divided into two subfamilies [23]. A previous study showed that *SlFBA4* is highly similar to *CpFBA* in *Arabidopsis* (*AtFBA1*/*2*/*3*), indicating that *SlFBA4* is likely located in the chloroplast where the CBC occurs [23]. Notably, we found that the expression of all eight FBAs was affected by low-temperature stress, especially *SlFBA4*, which were gradually upregulated by low-temperature stress from 0 day to three days [23]. This result indicates the possible roles of SlFBA4 in response to chilling stress. In addition, given that FBAs are key positive regulators of photosynthesis [39,40] and the fact that increasing the expression levels of *SlFBA7* resulted in enhanced chilling tolerance in tomato [24], we studied the functions of *SlFBA4* in tomato by creating over-expressed or silenced *SlFBA4* transgenic lines in tomato. 

In this study, we observed that an increase in FBA activity led to increases in seed and plant growth (Table 1) and Pn (Figure 3). Furthermore, we showed that a decrease in FBA activity led to an inhibitory effect on seed germination in optimal and sub-optimal temperature conditions (Table 1), indicating that *SlFBA4* is important for tomato growth and development, especially under low-temperature conditions. Similar results were also observed in other plants [41,42].

The root system is the most important structure to absorb water and nutrients, provide structural support and ensure tolerance against abiotic stresses [43,44]. Importantly, vigorous root system architecture leads to better growth and greater stress tolerance [40,44]. To our best knowledge, little is known about the role of FBAs in regulating root growth. In the present study, the promotion of root growth was clearly seen in over-expression transgenic lines and the roots look comparable between WT and RNAi lines (Figure 2A,B). It might be one of the reasons for the increased chilling stress tolerance of over-expression transgenic lines. It is worthy of further study to ascertain the role of *SlFBA4* in root growth. 

The adaptation of photosynthetic apparatus in cucumber seedlings to suboptimal conditions was related to the activation mechanisms of photosynthetic enzymes [45]. Both stomatal and nonstomatal factors influence photosynthesis in higher plants. Previous studies have shown that one of the most important nonstomatal factors is the activity of photosynthetic enzymes [45,46,47]. An increase in CBC enzyme activity may be the main factor that promotes the cucumber Pn increases during chilling conditions [48,49]. An increase in FBA activity promotes the reaction from dihydroxyacetone phosphate (DHAP) to fructose-1,6-diphosphate (FBP), and, as a consequence, the downstream reactions, including the reaction from FBP to fructose-6-phosphate (F6P) and a series of reactions to form RuBP, are also promoted. Increased RuBP regeneration could be the reason that Pn, growth and development increased. Low expression of *SlFBA4* in tomato leaves [23] was significantly enhanced in our over-expression transgenic plants (Figure 1A). Our results show that over-expression of *SlFBA4* led to higher CBC enzyme activity and Pn, whereas down-regulation of *SlFBA4* led to lower enzyme activity and Pn (Figure 3, Figure 6 and Figure 7). Many genetic engineering studies have confirmed that changes in CBC enzyme activity lead to changes in Pn and differences in chilling tolerance [23,50,51,52]. MDA, which is one of the best indexes of lipid peroxidation, may accumulate quickly in plant tissues and lead to DNA and biological membrane damage when exposed to chilling conditions [53]. However, our results show that *SlFBA4* over-expressing lines had a lower accumulation of MDA, whereas RNAi transgenic plants had a higher accumulation of MDA (Figure 8). These results indicate an important role of *SlFBA4* in protecting the cell membrane under chilling stress.

In conclusion, *SlFBA4* plays a crucial role in tomato growth and tolerance to chilling stress. Over-expression of *SlFBA4* enhanced photosynthetic capability in tomato and protected the cell membrane from MDA. Our findings shed new light on how CpFBA influences plant growth and development and regulates chilling tolerance. This provides a foundation to enhance plant adaptations to chilling conditions in solar greenhouses.

## 4. Materials and Methods

### 4.1. Plant Materials, Growth Conditions and Measurements

Seeds of the tomato inbred line ‘FF’ were used for germination tests and genetic transformation. The thousand-seed weight was measured using an analytical balance with 0.1 mg resolution. Germination tests were carried out in Petri dishes with moist filter paper at 28 °C and 18 °C in darkness. Seeds were grown in a solar greenhouse at Shandong Agriculture University in Tai’an under 25 °C/16 h days and 20 °C/8 h nights at a light intensity of 400 µmol m^−2^ s^−1^ and relative humidity of 75%. Seedlings at the five-true-leaf expansion stage were used for the following treatments. Plant height and stem diameter were measured using a ruler and a Vernier caliper, respectively. For the physiological parameters and analysis of enzyme activity, seedlings cultured in the greenhouse were used. For the low-temperature treatment, seedlings were transferred to a growth chamber at low temperature (8 °C days, 5 °C nights) and low light intensity (100 μmol m^−2^ s^−1^) for 0 h and 48 h. Fully expanded leaves were then harvested and frozen in liquid nitrogen. Samples were stored at −80 °C for subsequent experiments. Each experiment was repeated three times.

### 4.2. Gene Cloning, Vector Construction and Generation of Transgenic Lines 

To isolate the full-length coding sequence (CDS) for *SlFBA4*, young leaves were harvested from three-week-old seedlings and frozen immediately in liquid nitrogen. Total RNA was extracted using Trizol (Invitrogen, Solarbio Life Sciences, A208, Zicheng Pioneer Park, Xianghuangqi East Road, Haidian District, Beijing, China), and synthesis of cDNA was completed using the SuperScript^®^ First-Strand Synthesis System (Invitrogen) following the manufacturer’s instructions. The full-length CDS of *SlFBA4* was introduced into the plant expression vector pROKII driven by the CaMV-35S promoter. For RNA interference (RNAi) vector construction, two RNAi fragments with the same sequence (180 bp) and different restriction sites were cloned into pUCCRNAi to form a reverse-complementary hairpin structure as described previously [24]. Then, the hairpin structure was introduced into the plant expression vector pBI121 under control of the CaMV-35S promoter. 

For genetic transformation, the tomato inbred line ‘FF’ (obtained from Qinghua Shi, Shandong Agricultural University) was used. Positive plasmids were used for *Agrobacterium*-mediated tomato transformation as described by Fillatti et al. [54]. Regenerated shoots were screened on MS medium containing kanamycin (50 μg mL^−1^) and carbenicillin (300 μg mL^−1^). Resistant transformants that contained the over-expression plasmid were confirmed by polymerase chain reaction (PCR) using a 35S promoter forward primer and *SlFBA4* reverse primer (Appendix A). Resistant shoots containing the RNAi vector were also confirmed by PCR (Appendix A). The primary transformants (T0 generation) were self-fertilized in the solar greenhouse. The progeny obtained from T0 were named T1, and the progeny obtained from T1 were named T2. The T2 plants were used in the experiment.

### 4.3. Quantitative Reverse Transcription Polymerase Chain Reaction Analysis 

For analysis of gene expression, total RNA was isolated using Trizol according to the manufacturer’s instructions. First-strand cDNA was synthesized from 1 μg of total RNA with the PrimeScript 1st Strand cDNA Synthesis Kit (TaKaRa Bio Engineering Co., Ltd., Dalian, China). The relative mRNA expression from the transgenic plants was analyzed by quantitative reverse transcription PCR (qRT-PCR) using the TransStart TipTop Green qPCR SuperMix (Transgen, Yongtaizhuang North Road, Haidian District, Beijing, China), according to the manufacturer’s instructions. *β-actin* was used as an internal control. Amplification of targeted genes was performed using the LightCycler^®^ 480 II system (Roche, Penzberg, Germany). The analysis of relative mRNA expression data was performed using the 2^−ΔΔCt^ method [55]. qRT-PCR primers were designed to avoid conserved regions and to amplify 150 to 300 bp products (Appendix A). qRT-PCR was performed with three independent biological replicates.

### 4.4. Sodium Dodecyl Sulfate Polyacrylamide Gel Electrophoresis and Western Blot Analysis

Total protein from transgenic plants was extracted as previously described [56]. Approximately 1.0 g of clean leaf tissue was homogenized in 2 mL of extraction buffer (100 mM Tris-HCl (pH 8.0), containing 1 mM EDTA-Na_2_, 1% PVP, 10 mM mercaptoethanol and 0.2 M sucrose) followed by two centrifugations for 15 min at 4 °C and 12,000 rpm each. The liquid supernatant was stored at −80 °C and used for protein analysis. The polypeptide of the SlFBA4 protein was used for the polyclonal antibody preparation. The primary antibody was prepared by GenScript Biotechnology Co., Ltd. in Nanjing. Sodium dodecyl sulfate polyacrylamide gel electrophoresis (SDS-PAGE) and immunoblotting were carried out according to methods previously described [57]. Total protein was solubilized in 2 × SDS loading buffer and separated by SDS-PAGE using 12% separating gels and 5% concentrating gels containing 10% SDS. Proteins were then transferred to polyvinylidene fluoride membranes (Millipore, Burlington, MA, USA), which was followed by combination with the primary and secondary antibodies.

### 4.5. Measurement of Net Photosythetic Rate

The net photosynthetic rate (Pn) of expanded leaves was measured using a portable photosynthetic system (Ciras-3, PP Systems International, Hitchin, Hertfordshire, UK). Photon flux density (600 μmol m^−2^ s^−1^), CO_2_ concentration (380 mg L^−1^) and leaf temperature (25 °C) were maintained at a constant for all of the measurements. 

### 4.6. Enzyme Activity Assay

Ribulose-1,5-bisphosphate (RuBP) carboxylase (RuBPCase), fructose-1,6-bisphosphatase (FBPase), glyceraldehyde-3-phosphate dehydrogenase (GAPDH), sedoheptulose-1,7-bisphosphatase (SBPase), transketolase (TK) and FBA activities were determined as previously described [24,58]. 

### 4.7. Measurement of Malonaldehyde Content

Malonaldehyde (MDA) content was measured as described by Cho and Park in 2000 [59] and Heath and Packer in 1968 [60]. Briefly, approximately 0.5 g of fresh leaf tissue was prepared and homogenized in 4 mL of pre-cooling phosphate buffer. The liquid supernatant was stored at 4 °C for later use. Thiobarbituric acid colorimetry was used for measurement of MDA content. 

### 4.8. Statistical Analysis

Values are presented as means ± standard deviation of the three biological replicates. The data were statistically analyzed through the DPS software using one-way analysis of variance and Duncan’s multiple range test. Different lowercase letters indicate significant differences at *p* < 0.05.

## Figures and Tables

**Figure 1 ijms-23-00728-f001:**
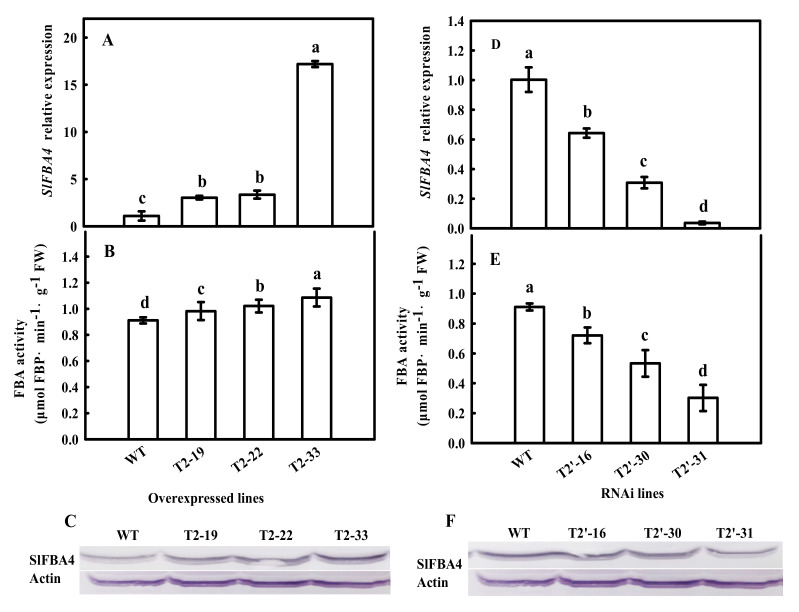
*SlFBA4* mRNA abundance, FBA activity and protein levels in WT and transgenic plants. (**A**,**D**) *SlFBA4* mRNA abundance in overexpression (**A**) and RNAi (**D**) plants, total RNA was separately isolated from the third fully expanded leaves of the WT and transgenic seedlings and subjected to real-time PCR analysis. (**B**,**E**) FBA activity, the same tissues used for *SlFBA4* mRNA analyses were sampled for FBA activity assay. (**C**,**F**) SlFBA4 protein levels, 25 μg of protein samples from leaves of WT and transgenic seedlings were separated by SDS-PAGE and polyclonal antibodies were used to detect FBA protein level. All values are presented as the mean ± SD (*n* = 3). Lowercase letters indicate that the mean values are significantly different among samples (*p* < 0.05).

**Figure 2 ijms-23-00728-f002:**
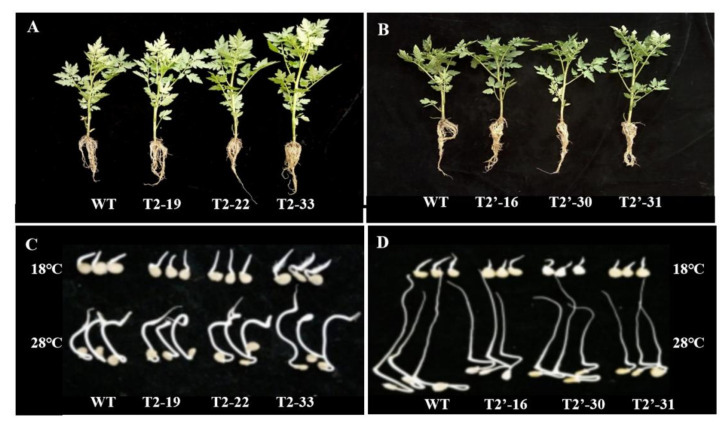
Seedlings and seed germination under different temperature conditions of the wild-type and transgenic lines of tomato. (**A**) Seedlings of WT and *SlFBA4* over-expression lines; (**B**) Seedlings of WT and RNAi lines; (**C**) Seed germination of WT and *SlFBA4* over-expression lines under 18 °C, and 28 °C; (**D**) Seed germination of WT and RNAi lines under 18 °C and 28 °C.

**Figure 3 ijms-23-00728-f003:**
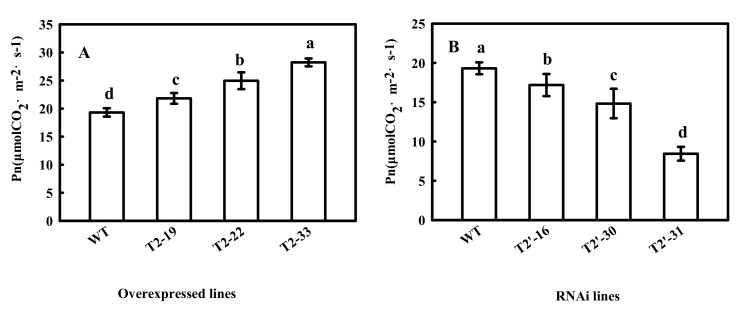
Net photosynthetic rate (Pn) in WT and transgenic tomato seedlings. Pn were measured at 25 °C under ambient carbon dioxide (CO_2_) (360 μmol mol^−1^). All values are presented as the means ± SD (*n* = 3). Lowercase letters indicate that the mean values are significantly different among the samples (*p* < 0.05).

**Figure 4 ijms-23-00728-f004:**
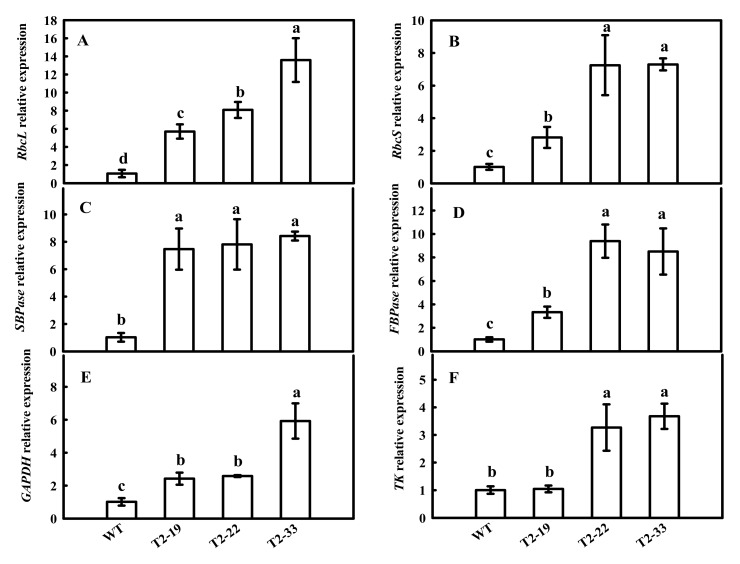
mRNA abundances of the six genes encoding important enzymes in the Calvin-Benson cycle in overexpressed *SlFBA4* transgenic tomato seedlings. Total RNA was separately isolated from the third fully expanded leaves of WT and transgenic plants and subjected to qRT-PCR analyses. (**A**) *rbcL*, (**B**) *rbcS*, (**C**) *SBPase*, (**D**) *FBPase*, (**E**) *GAPDH*, (**F**) *TK*. All values are presented as the means ± SD (*n* = 3). Lowercase letters indicate that the mean values are significantly different among the samples (*p* < 0.05).

**Figure 5 ijms-23-00728-f005:**
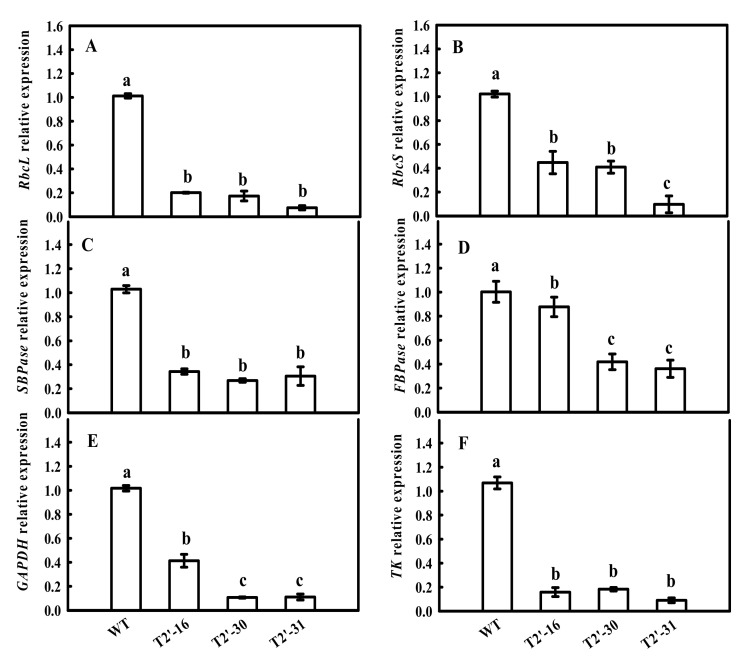
mRNA abundances of the six genes encoding important enzymes of the Calvin-Benson cycle in the RNAi seedlings using qRT-PCR. (**A**) *rbcL*, (**B**) *rbcS*, (**C**) *SBPase*, (**D**) *FBPase*, (**E**) *GAPDH*, (**F**) *TK*. All values are presented as the means ± SD (*n* = 3). Lowercase letters indicate that the mean values are significantly different among the samples (*p* < 0.05).

**Figure 6 ijms-23-00728-f006:**
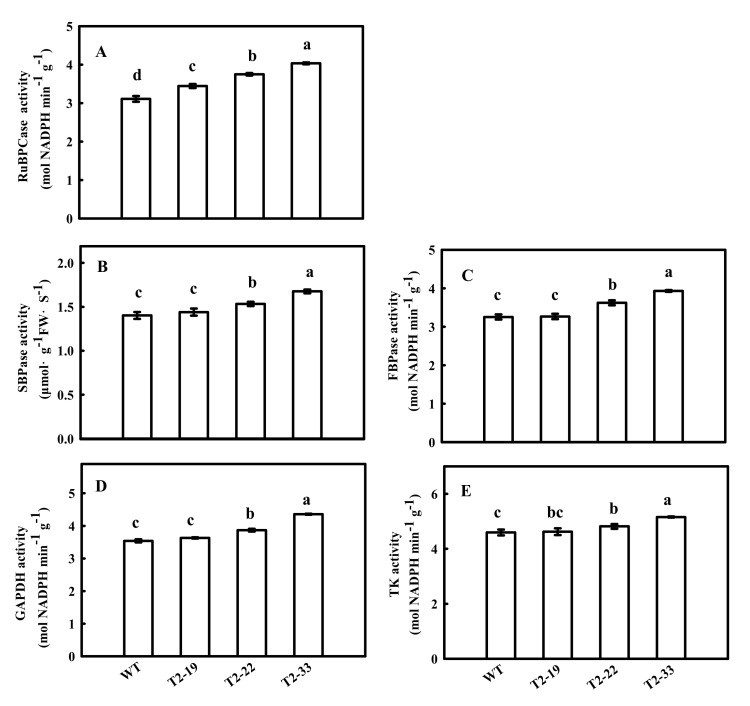
Enzyme activities of RuBPCase (**A**), SBPase (**B**), FBPase (**C**), GAPDH (**D**) and TK (**E**) in Calvin-Benson cycle in *SlFBA4* overexpressed transgenic tomato seedlings. The same tissues for mRNA analyses were sampled for the enzyme activities assay. All values are presented as the means ± SD (*n* = 3). Lowercase letters indicate that the mean values are significantly different among the samples (*p* < 0.05).

**Figure 7 ijms-23-00728-f007:**
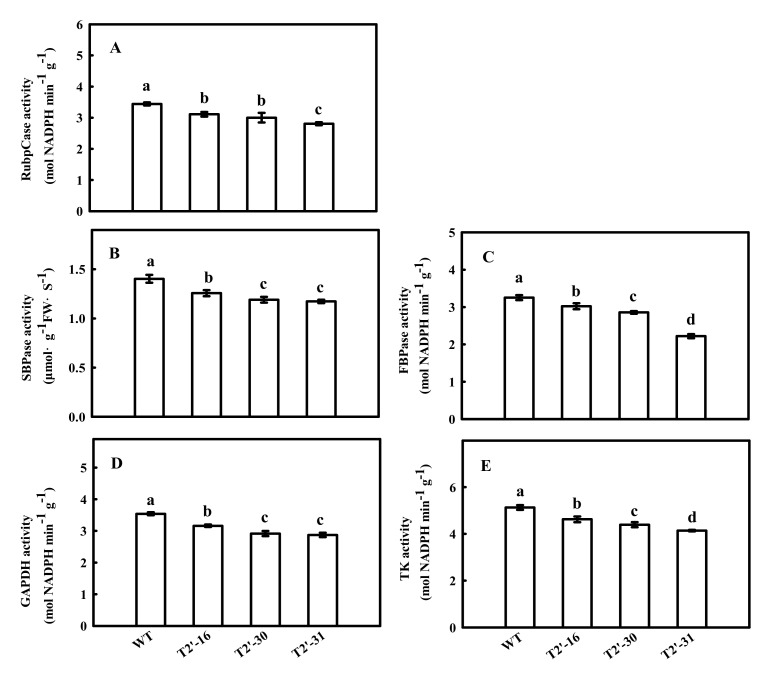
Enzyme activities of RuBPCase (**A**), SBPase (**B**), FBPase (**C**), GAPDH (**D**) and TK (**E**) in Calvin-Benson cycle in RNAi transgenic seedlings. All values are presented as the means ± SD (*n* = 3). Lowercase letters indicate that the mean values are significantly different among the samples (*p* < 0.05).

**Figure 8 ijms-23-00728-f008:**
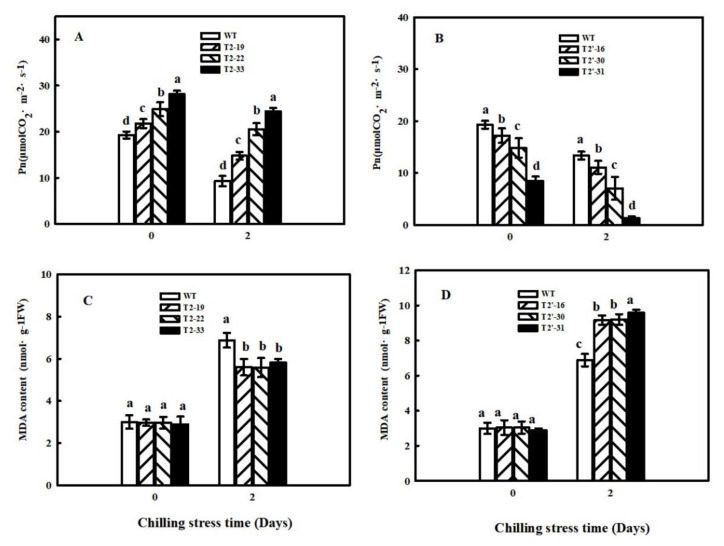
Effects of chilling stress on the net photosynthetic rate (**A**,**B**) and MDA content (**C**,**D**) in WT and transgenic lines. Seedlings were treated at 5 °C for 0 day and 2 days under 100 μmol m^−2^ s^−1^ PFD. After treatment, the third fully expanded leaves were used for the Pn and MDA content measurement. All values are presented as the means ± SD (*n* = 3). Lowercase letters indicate that the mean values are significantly different among the samples (*p* < 0.05).

**Table 1 ijms-23-00728-t001:** Comparison of morphological parameters of WT and transgenic plants.

Lines	Thousand-Seed-Weight	Germination Rate (%)		Plant Height (cm)	Stem Diameter (mm)
		18 °C	28 °C		
WT	2.2 ± 0.3 ^d^	37.3 ± 5 ^b^	97.0 ± 4 ^a^	25.4 ± 2 ^d^	14.2 ± 0.7 ^d^
T2-19	2.7 ± 0.2 ^c^	41.0 ± 3 ^ab^	96.6 ± 4 ^a^	29.0 ± 1 ^c^	16.5 ± 0.5 ^c^
T2-22	3.0 ± 0.2 ^b^	46.0 ± 5 ^a^	97.0 ± 3 ^a^	32.2 ± 1 ^b^	17.2 ± 0.3 ^b^
T2-33	3.2 ± 0.3 ^a^	48.6 ± 2 ^a^	99.3 ± 1 ^a^	35.4 ± 1 ^a^	21.0 ± 0.6 ^a^
T2′-16	2.1 ± 0.1 ^e^	34.3 ± 5 ^bc^	60.0 ± 2 ^b^	25.8 ± 1 ^d^	14.1 ± 0.7 ^d^
T2′-30	1.9 ± 0.2 ^f^	29.0 ± 1 ^cd^	53.7 ± 5 ^c^	26 ± 1 ^d^	13.7 ± 1.1 ^d^

Note: Values represent means ± SD; lowercase superscript letters indicate that the mean values are significantly different among samples (*p* < 0.05).

## Data Availability

Data is contained within the article or Appendix A.

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
