# Peer review of "Effects of the Chloroplast Fructose-1,6-Bisphosphate Aldolase Gene on Growth and Low-Temperature Tolerance of Tomato"

_ijms, 2022, doi:10.3390/ijms23020728_

Round 1

Reviewer 1 Report

Manuscript (MS) ID ijms-1502376 titled « Effects of the chloroplast fructose-1,6-bisphosphate aldolase gene on growth and low-temperature tolerance in tomato » aimed to decipher the role of the chloroplast fructose-1,6-bisphosphate aldolase in tomato development or subjected to chilling stress. Tomato (Solanum lycopersicum) is one of the most important crop grown worldwide but according to the authors, in Northen China, the crop is suffering of low temperature stress which hampering its production. Therefore, it is relevant to decipher the molecular mechanisms leading to tomato tolerance to this abiotic stress to improve its production in this part of the world. Based on the littérature review provided by the authors, Fructose-1,6-bisphosphate aldolase is involved in plant growth and responds to biotic and abiotic stresses that rising the need to investigate the role of this enzyme in tomato chilling stress tolerance. From this point of view, the study presented in this paper is relevant and the methodological approcahes are appropriate. Based on the results recorded in this paper, overexpression of Fructose-1,6-biphosphate aldolase gene led to plant growth and yield improvement in tomato, in contrast, silencing this gene showed an opposite effect. In chilling stress conditions, low level of malonaldehyde (MDA) content were observed in tomato plant overexpressing Fructose-1,6-biphosphate aldolase gene. Based on this, the authors concluded that this gene plays an important role in tomato growth and tolerance to chilling stress. Despite these significant findings, slight modifications are needed before this paper can be considered for publication in the « International Journal of Molecular Science ».

Specific comments

Please the attached PDF file.

reviews the state of the art on the role of light and brassinosteriod signals integration in plant development. For this purposis, the authors highlight the significant role played by light during seelding morphogenesis and by brassinosteroid during skotomorphogenesis. They also showed that light and brassinosteroid signals share a common regulatory components to promote plant growth and development. Despite this excellent state of the art reported by the authors, I think that the role of small RNA such as microRNA in the integration of light and brassinosteroid signals to promote plant growth and development can be reviewed. In conclusion, the paper needs slight modifications before it can be considered for publication in « International Journal of Molecular Sciences ».

Specific comments

Please see attached PDF document

Author Response

Response to Reviewer 1 Comments.

Point 1. Please improve the quality of the pictures

Response: The quality of the figures was improved.

Point 2. Please indicate 5' and 3' of the primers

Response: Corrected accordingly in the revised manuscript.

Point 3. Please indicate 5' and 3' of the primers (5'-3') and delete the ones at the begening and the end of the sequences

Response: Corrected accordingly in the revised manuscript.

Reviewer 2 Report

Insufficient resolution in Fig.S1 to illustrate the data presented.   It  is necessary to increase the digital resolution of the photos. 

In Fig.2. photos are presented of the seedling phenotype and seed germination under different temperature conditions of the wide type and transgenic lines of tomato. However, the text does not even describe the biometric data of the roots, especially since the differences are clearly visible.

It is desirable to expand the Discussion section.

There is negligence in the list of references, the year of issue is not always correctly set. The list of references should be carefully checked.

In the text, references to literature are not in brackets, but the authors and the year of issue are given.

Author Response

Response to Reviewer 2 comments.

Point 1. Insufficient resolution in Fig.S1 to illustrate the data presented. It is necessary to increase the digital resolution of the photos.

Response: The quality of the Fig.S1 was improved.

Point 2. In Fig.2. photos are presented of the seedling phenotype and seed germination under different temperature conditions of the wide type and transgenic lines of tomato. However, the text does not even describe the biometric data of the roots, especially since the differences are clearly visible.

Response: This is a very good point. Indeed, the promotion of root growth was clearly seen in over-expression transgenic lines and the roots looks comparable between WT and RNAi lines. However, since we didn’t perform statistical analysis of the root-related trait, we can not conclude whether SlFBA4 had significant effects on root growth. Your good point guides us to pay attention to the effects of SlFBA4 on root growth in future study, which might be one of the reasons for the increased chilling stress tolerance of over-expression transgenic lines. We also included this in the discussion section in the revised manuscript.

Point 3. It is desirable to expand the Discussion section.

Response: Thank you for your comments and the discussion section was expanded in the revised manuscript.

Point 4. There is negligence in the list of references, the year of issue is not always correctly set. The list of references should be carefully checked.

Response: Corrected accordingly in the revised manuscript.

Point 5. In the text, references to literature are not in brackets, but the authors and the year of issue are given.

Response: Corrected accordingly in the manuscript.

Reviewer 3 Report

I hope the following comments can help the authors in improving the quality of the manuscript:

Line 10: What is facility vegetables?

Line 15: Italicize the gene “SlFBA4”

Line 24: Break down the sentence to clearly specify the results of overexpression and RNAi silencing of the gene in “Moreover, an increase or decrease in thousand-seed weight, plant height, stem diameter and germination rate in optimal and sub-optimal temperatures was observed in the over-expression or RNAi lines, respectively”

Fig. S1. Label the marker size. If it is supplementary, it is not necessary to be included in the main text.

Fig 1. Which test was used for pairwise mean comparison? It is recommended that the lower error bars are also included in histograms.

Table 1: * is missing in the table, keep spacing between number and letters indicating significance or superscript them

Fig 2. Labels for seedlings are missing; notations and labels for genotypes are not clear

Fig. 3 CO2

Fig. 8. What does streatment mean?

Author Response

Response to Reviewer 3 Comments.

Point 1. Line 10: What is facility vegetables?

Response: The ‘facility’ was changed to ‘greenhouse’.

Point 2. Line 15: Italicize the gene “SlFBA4”

Response: Changed accordingly in the revised manuscript.

Point 3. Line 24: Break down the sentence to clearly specify the results of overexpression and RNAi silencing of the gene in “Moreover, an increase or decrease in thousand-seed weight, plant height, stem diameter and germination rate in optimal and sub-optimal temperatures was observed in the over-expression or RNAi lines, respectively”

Response: Thanks for your good suggestion. Corrected accordingly in the revised manuscript.

Point 4. Fig. S1. Label the marker size. If it is supplementary, it is not necessary to be included in the main text.

Response: Yes, you are right. The marker size was added and the Fig. S1 was removed to supplementary files.

Point 5. Fig 1. Which test was used for pairwise mean comparison? It is recommended that the lower error bars are also included in histograms.

Response: One-way ANOVA and Duncan’s multiple range test was used for statical analysis which was indicated in Statistical analysis section (line xx -line xx). Lower error bars were included in the revised manuscript.

Point 6. Table 1: * is missing in the table, keep spacing between number and letters indicating significance or superscript them

Response: This is a good point. The asterisk was supposed to be a notation, that’s why there was not * in the table. The * was changed to ‘Note:’. A space was added between number and letter, which indicate significant level.

Point 7. Fig 2. Labels for seedlings are missing; notations and labels for genotypes are not clear

Response: Labels and notations were added.

Point 8. Fig. 3 CO2

Response: CO2 was changed to CO2.

Point 9. Fig. 8. What does streatment mean?

Response: Sorry about the mistakes. It was changed to Chilling stress time.